# Early and Late Surgery-Free Rates of Conservative Management Strategy for Thrombosed Type A Acute Aortic Dissection and Acute Intramural Hematoma

**DOI:** 10.3390/jcm13185464

**Published:** 2024-09-14

**Authors:** Kiyotoshi Akita, Yoshiyuki Takami, Atsuo Maekawa, Koji Yamana, Kentaro Amano, Kazuki Matsuhashi, Wakana Niwa, Yasushi Takagi

**Affiliations:** Department of Cardiovascular Surgery, Fujita Health University School of Medicine, Toyoake 470-1192, Aichi, Japan; k-akita@fujita-hu.ac.jp (K.A.);

**Keywords:** type A acute aortic dissection, aortic intramural hematoma, retrograde thrombosed type A acute aortic dissection

## Abstract

**Background:** We have employed a conservative management approach, including intensive control of both blood pressure and heart rate, in patients with aortic intramural hematoma (AIMH) and retrograde thrombosed type A acute aortic dissection (RT-TAAAD), sharing common clinical and imaging characteristics. **Methods:** To evaluate the outcomes of our conservative management approach, we retrospectively reviewed the clinical records of 98 patients diagnosed with AIMH or RT-TAAAD from January 2008 to March 2023. A conservative management approach was applied, except for those patients with an aortic diameter ≥ 55 mm, false lumen expansion, or cardiac tamponade, who underwent emergency aortic repair. **Results:** Besides 2 patients, who declined surgery and subsequently died from aortic rupture, 18 patients underwent urgent aortic surgery, while 78 did not. Multivariable logistic regression analysis identified the extrusion type of ulcer-like projections (ULPs) on admission and a maximum aortic diameter ≥ 45 mm on Day 1 as risk factors for acute aortic surgery. Among the 78 patients who were discharged, 9 (12%) underwent aortic surgery, while 69 (88%) did not, with a median follow-up of 44 months. The overall actuarial aortic surgery-free rates were 78% at 1 year and 72% at 5 years, respectively. A Cox proportional hazards analysis identified ULPs and an aortic diameter ≥ 45 mm at discharge as risk factors for late aortic surgery. **Conclusions:** The early and late outcomes of our conservative strategy for AIMH and RT-TAAAD demonstrate favorable surgery-free rates. The extrusion type of ULPs on admission and an aortic diameter ≥ 45 mm on Day 1 are predictors of acute aortic surgery, while ULPs and an aortic diameter ≥ 45 mm at discharge are predictors of late surgery.

## 1. Introduction

A thrombosed false lumen in the ascending aorta in a type A acute aortic dissection (TAAAD) is a condition distinct from a classic TAAAD, which is highly fatal without emergency surgery [1]. This specific disorder of TAAAD includes aortic intramural hematoma (AIMH) and retrograde thrombosed type A acute aortic dissection (RT-TAAAD). Although their pathological processes differ, AIMH and RT-TAAAD share common clinical behavior and imaging characteristics. These are characterized by the presence of a hematoma within the aortic wall without an entry tear in the ascending aorta, and without direct communication between the true and false lumen [1,2]. Although they are recognized as a crescent or circular local aortic wall thickening, they sometimes accompany an ulcer-like projection (ULP) [3], a deep and ulcerated lesion or a localized blood-filled pouch protrusion in the thickest part of the intramural hematoma, without any visible intimal flap.

Patient management in this scenario remains controversial. While the management of AIMH and RT-TAAAD has mostly involved emergency or urgent surgery in the United States and Europe, there has been an upward trend in conservative patient management without surgery in Asian countries [2,4]. We have employed a conservative management approach through the intensive control of both blood pressure and heart rate in Japan. Even with a conservative management approach, AIMH and RT-TAAAD represent a dynamic entity associated with a progression to life-threatening TAAAD, aortic expansion or rupture, malperfusion or organ ischemia, or developing a flow communication between the true and false lumen, all of which require surgical intervention [3].

In the present study, we retrospectively investigated the early and late outcomes of our conservative approach for patients with AIMH and RT-TAAAD in daily practice, aiming to identify the factors associated with surgery-free rates in these patients.

## 2. Materials and Methods

### 2.1. Ethical Statement

This retrospective study, which analyzes single-center data from our routine medical practice for patients with acute aortic dissection, was comprehensively approved by the institutional review board; a waiver of informed consent was included (HM19-323, 15 October 2019). It was conducted in accordance with the ethical guidelines for clinical studies published by the Ministry of Health and the Helsinki Declaration.

### 2.2. Study Patients

From January 2008 to March 2023, 98 consecutive patients at our institution were diagnosed with AIMH or RT-TAAAD within 24 h of admission. Diagnoses were established by cardiologists and certified radiologists using contrast-enhanced computed tomography (CT) with a 64-row multidetector scanner. Plain images were taken before injecting the contrast medium; early-phase images were taken with contrast enhancement; and delayed-phase images taken 150 s after the injection of the contrast medium were obtained.

AIMH and RT-TAAAD were diagnosed as thrombosed false lumen or non-communicating aortic dissection [2], defined as follows in either early- or delayed-phase CT images: (1) presence of a crescent-shaped false lumen, or (2) no tear or blood flow entering from a tear in the ascending aorta, according to the Japanese guideline [5]. RT-TAAAD was specifically characterized by an intimal tear in the descending aorta or the abdominal aorta with a retrograde extension of the thrombosed false lumen in the ascending aorta. ULPs were defined as small protrusions (the contrast region of <15 mm in the cephalocaudal direction [6]) in a thrombosed false lumen on contrast-enhanced CT scans.

### 2.3. Patient Management

Once AIMH and RT-TAAAD were diagnosed, and the thrombosed false lumen in the ascending aorta was confirmed, conservative patient management with strict control of blood pressure (systolic blood pressure < 120 mmHg) and heart rate (<70 beats per minute) was started, even if the ascending aortic diameter was >50 mm and/or the false lumen thickness was >11 mm. The patients were treated nonoperatively in accordance with our treatment protocol and rehabilitation program (Table 1).

On Day 0, patients fasted and remained on bed rest in the intensive care unit (ICU). A CT scan with or without contrast enhancement was repeated on Days 1, 3, 5, 7, 14, 21, and 28, depending on the patient’s kidney function. Based on the absence of exacerbation in CT findings, the following activities were allowed: On Day 3, the sitting position was allowed, oral intake commenced, and intravenous antihypertensives were gradually switched to oral medications. On Day 7, toileting was allowed. On Day 14, patients started walking in the ward and using the shower. Patients were discharged home after Day 28.

During this conservative management approach, a contrast-enhanced CT scan was also performed if the patient developed any symptoms. Emergency aortic repair was indicated for the following reasons: (1) aortic diameter enlargement ≥ 55 mm, (2) false lumen expansion with ULPs, (3) false lumen recanalization, (4) cardiac tamponade or increased pericardial effusion, or (5) uncontrolled hypertension and pain. The extent of graft replacement was decided based on the location of the primary entry. Total arch replacement with a frozen elephant trunk was indicated when the primary tear existed at the distal aortic arch, or when the false lumen of the descending aorta was patent.

### 2.4. Data Collection and Study Outcomes

The medical records of each study patient were retrospectively reviewed for demographics at the time of presentation, CT scan findings, and outcomes. The focused findings on CT scans included the maximum aortic diameter, the maximum false lumen diameter, and the presence or absence of ULPs in the ascending aorta.

When an ULP was present, it was classified as a protrusion or extrusion, modeled after the definition of herniated discs [7], as shown in Figure 1. A protrusion type of ULP was identified if the greatest length of the intimal defect at the ulcer site (a) was greater than the width of the intramural blood pool (b) in any plane. An extrusion type of ULP was identified when the length of (a) was less than the width of (b).

The primary early outcome was freedom from thoracic aortic repair during the hospital stay after the initial onset of AIMH and RT-TAAAD. The primary late outcome was a thoracic aortic surgery-free survival rate. A follow-up was performed during patient visits or by telephone. The study was closed on 30 September 2023.

### 2.5. Statistical Analysis

Categorical data are represented as either a number or a percentage (%); continuous data are written in the form of a mean and a standard deviation, or as a median and an interquartile range (IQR) when the distribution was skewed after evaluation of normality by the Shapiro–Wilk test. Comparisons between patients undergoing aortic repair during their hospital stay and those who did not undergo aortic repair were conducted using the Mann–Whitney U test for continuous variables and Fisher’s exact test for categorical variables. Receiver operating characteristic (ROC) curves were generated to determine the cut-off values for predicting aortic surgery in both the acute and chronic phases.

To identify independent predictors of acute aortic surgery during the hospital stay, multivariable logistic regression analysis was performed using selected explanatory variables. For the selection of these explanatory variables, the variance inflation factor was 10 or less, as determined by a multicollinearity test.

Long-term thoracic aortic surgery-free rates were estimated using the Kaplan–Meier method. A multivariable Cox proportional hazards analysis with a backward stepwise method was used to identify risk factors for aortic surgery in the chronic phase. A competing risk analysis using the Gray test was performed for late aortic surgery, as non-aortic death could be a confounding factor using traditional methods. We plotted log (time) versus log (2log[survival]), stratified by each significant risk factor, and evaluated whether the plotted lines were parallel. All statistical analyses were performed using software (EZR, version 1.65, available on the website https://www.jichi.ac.jp/saitama-sct/SaitamaHP.files/statmed.html, accessed on 1 August 2024 [8]). Any *p* value less than 0.05 was considered statistically significant.

## 3. Results

### 3.1. Early Outcomes

As shown in Figure 2, of the 98 study patients diagnosed with AIMH or RT-TAAAD, 18 (18%) underwent aortic surgery during their hospital stay due to false lumen recanalization (n = 8, 44%), false lumen expansion with ULP (n = 6, 33%), aortic diameter enlargement >55 mm (n = 3, 17%), and increased pericardial effusion (n = 1, 6%). All 18 patients who underwent surgery survived and were discharged from our hospital. Three patients with surgical indications declined aortic surgery due to advanced age; two of these patients subsequently died from aortic rupture.

Comparing characteristics between those patients who underwent aortic surgery during a hospital stay (n = 18) and those patients who did not (n = 78), there were significantly more patients undergoing surgery with an ULP on admission (78% vs. 41%, *p* = 0.008), and a significantly larger maximum aortic diameter on Day 1 (45 mm [IQR: 44, 49] vs. 43 mm [IQR: 41, 46], *p* = 0.032), although basic demographics were similar (Table 2). Specifically, the extrusion type of an ULP on admission was significantly more common than the protrusion type in patients undergoing acute aortic surgery (88% vs. 12%, *p* < 0.001).

Multivariable logistic regression analysis identified an extrusion type of ULP on admission and a maximum aortic diameter on Day 1 as independent risk factors for acute aortic surgery (Table 3). An ROC analysis revealed that a maximum aortic diameter ≥ 45 mm on Day 1 was the optimal cut-off value for predicting acute aortic surgery during the hospital stay, with a sensitivity of 92%, a specificity of 46%, and an area under the curve of 0.703 (95% CI: 0.571–0.836).

### 3.2. Late Outcomes

Follow-up data were available for all 78 patients who were discharged from our hospital (100% complete). The median duration for a follow-up was 45 months (IQR: 18, 78); the average duration was 53 ± 42 months. Of these 78 patients, 9 (12%) underwent aortic surgery due to aortic aneurysmal enlargement at a median of 17 months (IQR: 6, 60) after discharge, while 69 (88%) did not. Among those patients who did not undergo surgery, two died of an aortic aneurysm at 84 and 101 months after discharge, respectively. The causes of non-aortic deaths included cancer (n = 4), an infection (n = 4), a stroke (n = 1), and a hematological disorder (n = 1). The overall actuarial aortic surgery-free rates were 80.4% at 6 months, 77.9% at 1 year, 71.7% at 5 years, and 60.3% at 10 years, respectively (Figure 3). A Competing-risk analysis, considering non-aortic death as a competing risk, was performed using the method of Fine and Gray (*p* = 0.001, Gray test) (Figure 3). The cumulative incidence estimates for non-aortic death were 5.3% at 6 months, 7.1% at 1 year, 11.2% at 5 years, and 17.4% at 10 years, respectively.

Comparing patient characteristics between those who underwent late aortic surgery and those who did not, the maximum aortic diameters on admission, on Day 1, and at discharge were significantly larger in patients who underwent surgery (Table 4). In addition, an ULP at discharge was significantly more prevalent in patients who underwent late surgery compared to those who did not (67% vs. 17%, *p* = 0.004). An ROC analysis revealed that a maximum aortic diameter ≥ 45 mm at discharge was the optimal cut-off value for predicting late aortic surgery at follow-up, with a sensitivity of 89%, a specificity of 63%, and an area under the curve of 0.794 (95% CI: 0.624–0.964). A Cox proportional hazards analysis identified an ULP at discharge as the independent risk factor for late aortic surgery (Table 5).

## 4. Discussion

The present study, which focused on thrombosed TAAAD including aortic AIMH and RT-TAAAD, yielded the following main findings: (i) Aortic diameter and false lumen diameter on admission did not predict the urgent need for aortic surgery during the hospital stay. Instead, the extrusion type of ULPs on admission and a maximum aortic diameter ≥ 45 mm on Day 1 were predictors of acute aortic surgery for AIMH and RT-TAAAD. (ii) An extrusion type of an ULP, in particular, was a predictor of acute aortic surgery for AIMH and RT-TAAAD, whereas a protrusion type of an ULP was less likely to predict acute aortic surgery. (iii) The overall actuarial aortic surgery-free rates were 77.9% at 1 year, 71.7% at 5 years, and 60.3% at 10 years, respectively. (iv) A maximum aortic diameter ≥ 45 mm at discharge and the presence of ULPs at discharge were predictors of late aortic surgery at follow-up.

Our first finding, that aortic diameter and false lumen diameter on admission did not predict acute aortic surgery, may be opposed to the 2020 Japanese [5] and 2022 ACC/AHA [9] guidelines suggesting initial medical management only for high-risk patients with the maximum aortic diameter from < 40 to 50 mm, and a hematoma thickness of <10 mm. Both American and Japanese guidelines recommend prompt open surgical repair in patients with an uncomplicated AIMH as class IIa, with an evidence level of C. Therefore, this strategy will need to be further validated before it can be adopted in daily practice. According to our results, CT findings on Day 1, rather than on admission, are more predictive. Our finding that a maximum aortic diameter ≥ 45 mm on Day 1 predicted acute aortic surgery may be attributed to the dynamic changes in the thrombosed false lumen. Therefore, careful observation with frequent contrast-enhanced CT scans, along with strict control of blood pressure and heart rate, is crucial in the conservative management strategy for AIMH and RT-TAAAD.

The dynamic changes observed in thrombosed TAAAD may be related to the following pathophysiology. In RT-TAAAD, the absence of an intimal defect and flow communication between the true and false lumens are considered essential for diagnosing AIMH. This may result from the rupture of the vasa vasorum and hemorrhage in the medial layer of the aorta [10,11]. However, focal intimal disruptions (FIDs) are frequently observed on multidetector CT imaging [12]. Approximately two-thirds of the patients with type B AIMH had FIDs at admission [13]. Therefore, an AIMH may partially originate from an FID as a primary entry tear similar to classic aortic dissection, whereas an RT-TAAAD forms closed and thrombosed false lumens due to the lack of a re-entry tear [14].

Our second finding, that the protruding type of ULPs predicted acute aortic surgery for AIMH and RT-TAAAD may be unique. Although there seems to be common pathophysiology, an ULP, which is one of the CT findings of aortic dissection, differs from a PAU, which is defined as an ulceration of the arteriosclerotic lesions extending below the internal elastic lamina and media [6], with a pathogenesis different from aortic dissection. Most previous reports recommend timely acute aortic surgery in the presence of an ULP and its enlargement due to poor prognosis [1,9,15,16]. However, previous studies have also suggested that the size of the FID or the ULP is highly associated with aorta-related events [13,17]. To our knowledge, no studies have investigated the association between the shape of an ULP and the clinical outcomes in thrombosed TAAAD. Our results emphasize the clinical importance of the protruding type of ULPs.

Our third finding regarding the overall actuarial aortic surgery-free rates supports the conservative management strategy for thrombosed TAAAD. This approach, which has shown favorable outcomes mostly in East Asia, including Japan [18,19,20], contrasts with the 2022 ACC/AHA guideline recommendations for emergency open surgical intervention [9]. On the other hand, several reports from Asia have indicated favorable survival rates after surgery, even in elderly patients with thrombosed TAAAD, due to an increased risk of cardiorespiratory complications during conservative management [21,22]. A recent meta-analysis concluded that surgery was associated with better late survival and lifetime gain in comparison with medical therapy alone, even though all the identified studies turned out to be of Asian origin [4]. In addition, a recent report suggests that surgical management of AIMH may yield excellent short- and mid-term survival when compared to classic TAAAD [23]. Patients with AIMH may not be as sick as patients in the classic TAAAD group, due to fewer severe aortic insufficiencies and malperfusion. The extent of aortic repair for AIMH may be less than that for the classic TAAAD group, such as fewer aortic root replacements and arch replacements. Therefore, our conservative management strategy should be validated further.

Our fourth finding, that a maximum aortic diameter ≥ 45 mm and the presence of an ULP at discharge predict late aortic surgery, may also be considered novel. To our knowledge, no studies have identified predictors for late surgery. Chen et al. [1] reported that 100% of ULP enlargements were found during follow-up over 5 years in patients with thrombosed TAAAD, and that 57.1% of late aortic deaths occurred if PAUs were untreated, although small ULPs might not be initially visible. As demonstrated in our study, the presence of an ULP at discharge is a significant factor for subsequent surgery to prevent aortic death during follow-up.

The present study has several limitations. First, it is a single-center, retrospective observational study with a small number of patients, and it lacks a control group for comparison. Second, there may be bias regarding CT characteristics due to advancements in imaging technology during the study period. Third, refractory pain was not investigated, although it is considered a risk factor for rupture [20,24]. Including subjective pain scales as a criterion for managing a life-threatening disease may be inappropriate. Fourth, we need further investigation of sex differences in the early and late outcomes of our conservative management strategy for AIMH and RT-TAAAD; sex difference is an important aspect in outcomes of acute aortic dissection [25].

## 5. Conclusions

The early and late outcomes of our conservative management strategy for AIMH and RT-TAAAD demonstrate favorable surgery-free rates. The extrusion type of ULPs on admission and an aortic diameter ≥ 45 mm on Day 1 are predictors of the need for acute aortic surgery, while the presence of ULPs and an aortic diameter ≥ 45 mm at discharge are predictors of late surgery.

## Figures and Tables

**Figure 1 jcm-13-05464-f001:**
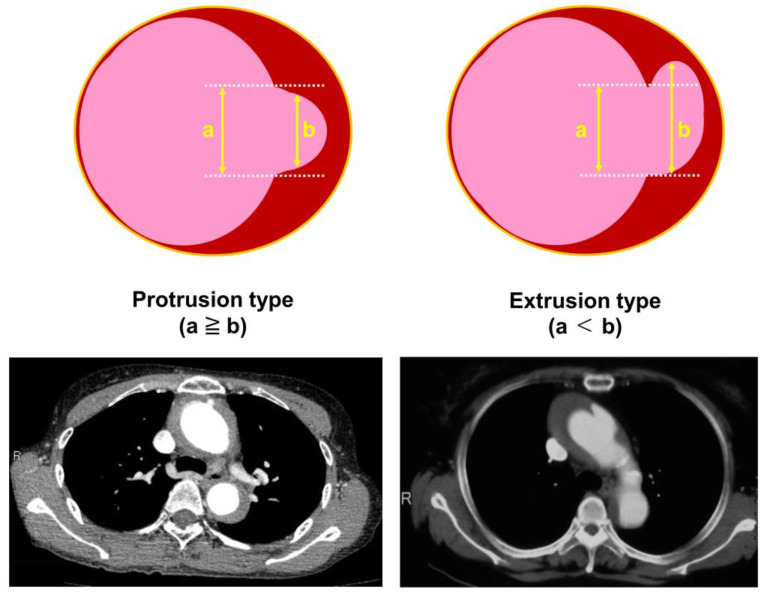
Types of ulcer-like projections (ULPs) associated with thrombosed type A acute aortic dissection and intramural hematoma. A protrusion type ULP is present when a ≥ b, while an extrusion type ULP is present when a < b. a: length of intimal defect at ulcer site; b: width of intramural blood pool.

**Figure 2 jcm-13-05464-f002:**
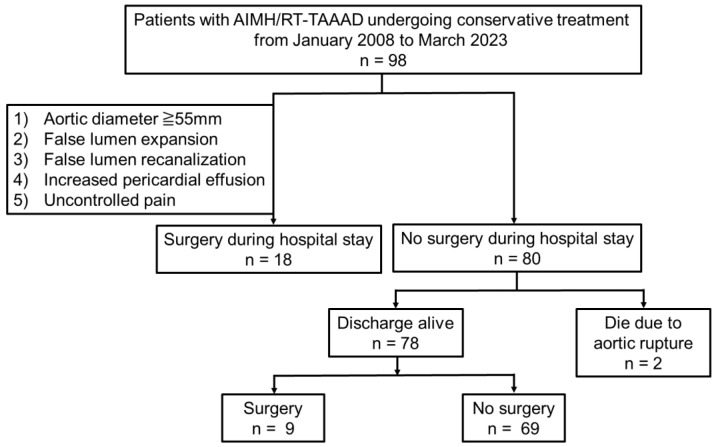
Flowchart showing the inclusion criteria for study patients with aortic intramural hematoma (AIMH) and retrograde thrombosed type A acute aortic dissection (RT-TAAAD), as well as the distribution of patients who underwent acute aortic surgery during their hospital stay versus those who underwent surgery in the chronic phase.

**Figure 3 jcm-13-05464-f003:**
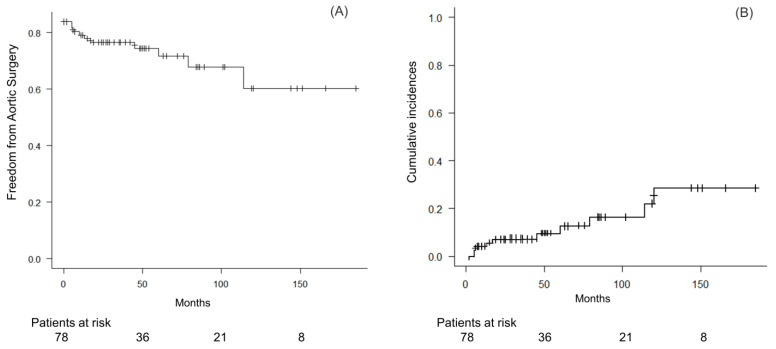
(**A**) Aortic surgery-free survival rates. (**B**) Cumulative incidence of aortic surgery, with non-aortic death as a competing risk.

**Table 1 jcm-13-05464-t001:** Treatment protocol for thrombosed type A acute aortic dissection and intramural hematoma.

	Date of Onset	Day 1	Day 3	Day 5	Day 7	Day 14	Day 21	Day28
Contrast-enhanced CT	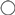	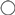	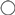	△	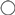	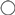	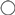	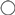
Rest	Bed rest	Sitting position	Toilet	Walking	Discharge
insidethe ward	insidethe hospital
Meal	Fasting & infusion	Oral intake

CT, computed tomography; 
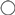
, CT is performed; △, CT is performed if the CT finding on Day 3 was significant.

**Table 2 jcm-13-05464-t002:** Comparison between the patients who did and did not undergo aortic surgery during hospital stay.

	Overall n = 98	Declined Surgery n = 2	Surgery n = 18	No Surgery n = 78	*p* Value
Demographics					
Age, years	73 [65, 81]	78 [76, 79]	69 [64, 80]	73 [65, 81]	0.507
Female, n (%)	55 (56)	2 (100)	11 (61)	42 (54)	0.192
Smoking, n (%)	30 (31)	1 (50)	5 (28)	24 (31)	0.234
Hypertension, n (%)	96 (98)	2 (100)	18 (100)	76 (97)	1.000
Hyperlipidemia, n (%)	22 (22)	1 (50)	1 (9)	20 (26)	0.110
Diabetes mellitus, n (%)	3 (3)	0	0	3 (3)	1.000
CKD, n (%)	18 (18)	0	3 (17)	15 (19)	1.000
CT findings on admission					
Aortic diameter, mm	45 [42, 48]	47 [45, 50]	46 [44, 49]	45 [42, 48]	0.308
Thickness of thrombosed FL or IMH, mm	12 [8, 15]	12 [11, 12]	11 [9, 15]	12 [8, 15]	0.649
Presence of ULPs, n (%)	48 (49)	2 (100)	14 (78)	32 (41)	0.008
protrusion type, n (%)	32 (67)	1 (50)	3 (21)	28 (88)	<0.001
extrusion type, n (%)	16 (33)	1 (50)	11 (79)	4 (12)	
CT findings on Day 1					
Aortic diameter, mm	44 [42, 46]	48 [46, 49]	45 [44, 49]	43 [41, 46]	0.032
Thickness of thrombosed FL or IMH, mm	8 [6, 9]	13 [13, 14]	8 [6, 9]	8 [6, 9]	0.744

Data are expressed as medians [interquartile ranges] or as numbers (%). Abbreviations: CKD, chronic kidney disease; CT, computed tomography; FL, false lumen; IMH, intramural hematoma; ULP, ulcer-like projection.

**Table 3 jcm-13-05464-t003:** Univariable and multivariable analysis for predictors of acute aortic surgery.

Variables	Univariable		Multivariable	
	HR (95% CI)	*p* Value	HR (95% CI)	*p* Value
Patient Demographics				
Age	0.99 (0.938–1.042)	0.719		
Female	2.68 (0.784–9.153)	0.116		
Hyperlipidemia	0.25 (0.031–2.060)	0.198		
CT findings on admission				
Aortic diameter	1.07 (0.943–1.213)	0.291		
Thickness of thrombosed FL or IMH	1.05 (0.923–1.194)	0.464		
Presence of ULPs	6.27 (1.632–24.10)	0.008	3.62 (0.614–21.40)	0.155
Protrusion type of ULP	0.49 (0.127–1.893)	0.301		
Extrusion type of ULP	27.4 (6.472–116.2)	<0.001	6.91 (1.193–40.10)	0.031
CT findings on Day 1				
Aortic diameter	1.15 (1.014–1.322)	0.036	1.62 (1.084–3.046)	0.038
Thickness of thrombosed FL or IMH	0.93 (0.774–1.134)	0.475		

Abbreviations: HR, hazard ratio; CI, confidence interval; CT, computed tomography; FL, false lumen; IMH, intramural hematoma; ULP, ulcer-like projection.

**Table 4 jcm-13-05464-t004:** Comparison between the patients who did and did not undergo aortic surgery at follow-up.

	Surgery n = 9	No Surgery n = 69	*p* Value
Demographics			
Age, years	68 [62, 75]	73 [65, 81]	
Female, n (%)	5 (56)	34 (49)	1.000
Hypertension, n (%)	9 (100)	66 (96)	1.000
Hyperlipidemia, n (%)	3 (33)	14 (20)	0.400
Diabetes mellitus, n (%)	0	3 (4)	1.000
CKD, n (%)	2 (22)	12 (17)	0.660
CT findings on admission			
Maximum aortic diameter, mm	48 [46, 52]	44 [42, 47]	0.008
Maximum thickness of thrombosed FL or IMH, mm	12 [10, 15]	11 [7, 14]	0.214
Presence of ULPs, n (%)	5 (56)	25 (36)	0.294
protrusion type, n (%)	3 (60)	23 (92)	0.119
extrusion type, n (%)	2 (40)	2 (8)	
CT findings on Day 1			
Maximum aortic diameter, mm	45 [43, 52]	43 [41, 46]	0.021
Maximum thickness of thrombosed FL or IMH, mm	7 [6, 13]	8 [6, 9]	0.612
CT findings on discharge			
Maximum aortic diameter, mm	47 [43, 50]	42 [38, 45]	0.004
Maximum thickness of thrombosed FL or IMH, mm	6 [4, 10]	3 [2, 5]	0.034
Presence of ULPs, n (%)	6 (67)	12 (17)	0.004
protrusion type, n (%)	3 (33)	10 (14)	0.167
extrusion type, n (%)	3 (33)	2 (3)	0.009

Data are expressed as medians [interquartile ranges] or as numbers (%). Abbreviations: CKD, chronic kidney disease; CT, computed tomography; FL, false lumen; IMH, intramural hematoma; ULP, ulcer-like projection.

**Table 5 jcm-13-05464-t005:** Univariable and multivariable analysis for predictors of late aortic surgery.

Variables	Univariable		Multivariable	
	HR (95% CI)	*p* Value	HR (95% CI)	*p* Value
CT findings on Day 1				
Aortic diameter	1.20 (1.062–1.353)	0.003	0.85 (0.587–1.243)	0.411
Thickness of thrombosed FL or IMH	1.041 (0.891–1.217)	0.613		
CT findings on discharge				
Aortic diameter	1.28 (1.123–1.473)	<0.001	2.62 (1.884–6.076)	0.016
Thickness of thrombosed FL or IMH	1.08 (0.963–1.212)	0.189	0.91 (0.698–1.187)	0.487
Extrusion type of ULP	7.68 (1.892–11.25)	0.004	4.75 (1.478–8.631)	0.024

## Data Availability

The raw data supporting the conclusions of this article will be made available by the authors on request.

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
