# Peer review of "Early and Late Surgery-Free Rates of Conservative Management Strategy for Thrombosed Type A Acute Aortic Dissection and Acute Intramural Hematoma"

_jcm, 2024, doi:10.3390/jcm13185464_

Round 1

Reviewer 1 Report

Comments and Suggestions for Authors

We thank the authors for submitting their manuscript to our journal. The article focuses on the outcomes of conservative strategies in cases of thrombosed acute Type A aortic dissection and intramural hematoma. The results are well-presented, and the conclusions are supported by the findings. However, the following revisions are required:

1) The Background section of the abstract needs to be reformulated, as it currently describes the methods rather than providing contextual information.

2) In the Methods section, please specify the average duration of follow-up. Additionally, clarity regarding patient retention and reasons for loss to follow-up would enhance the reliability of the reported long-term outcomes.

3) Include smoking habits and pharmacological therapy in the descriptive variables of the sample tables.   

4) Discuss the results in light of the following study:

Ahmad, Rana-Armaghan et al. “Acute type A intramural hematoma: The less-deadly acute aortic syndrome?.” The Journal of Thoracic and Cardiovascular Surgery, S0022-5223(24)00090-4. 25 Jan. 2024, doi:10.1016/j.jtcvs.2024.01.032.  

5) Expand the discussion of the results in the context of existing guidelines and consensus documents on the subject.

6) It is recommended to create a graphical abstract to convey the main message of the study effectively.

7) Highlight any sex-related differences in the outcomes considered.

With these revisions, the manuscript will provide a more comprehensive understanding of the conservative strategy's effectiveness in treating acute Type A aortic dissection and intramural hematoma.

Author Response

Thank you very much for taking the time to review this manuscript. Please find the detailed responses below and the corresponding revisions highlighted in red in the re-submitted files. We also appreciate your appropriate understanding of our article focusing on the outcomes of conservative strategies in cases of thrombosed acute Type A aortic dissection and intramural hematoma.

Comments 1: The Background section of the abstract needs to be reformulated, as it currently describes the methods rather than providing contextual information.

Response 1: Thank you for pointing this out. However, we do not agree with this comment. The Background section of our abstract provides the contextual information that we have employed conservative management by intensive control of both blood pressure and heart rate in the patients with aortic intramural hematoma and retrograde thrombosed type A acute aortic dissection. Also, Reviewer 2 commented that the introduction provide sufficient background and include all relevant references.

Comments 2: In the Methods section, please specify the average duration of follow-up. Additionally, clarity regarding patient retention and reasons for loss to follow-up would enhance the reliability of the reported long-term outcomes.

Response 2: Thank you for pointing this out. We agree with this comment. According to the reviewer’s recommendation, we have added the average duration of follow-up in the section of Late outcomes in Results (Lines 165 to 167). The follow-up rate was 100%, enhancing the reliability of our reported long-term outcomes, as the reviewer suggested.

Comments 3: Include smoking habits and pharmacological therapy in the descriptive variables of the sample tables.  

Response 3: As the reviewer recommend, we have added smoking habits in the revised Table 2, describing the demographics of the study patients. However, we do not include the pharmacological therapy, because it was similar in all the study patients to control blood pressure and heart rate.

Comments 4: Discuss the results in light of the following study:

Ahmad, Rana-Armaghan et al. “Acute type A intramural hematoma: The less-deadly acute aortic syndrome?.” The Journal of Thoracic and Cardiovascular Surgery, S0022-5223(24)00090-4. 25 Jan. 2024, doi:10.1016/j.jtcvs.2024.01.032.   

Response 4: We appreciate your kind suggestion for me to pro and con discussion about our conservative strategies in cases of thrombosed acute Type A aortic dissection and intramural hematoma, referring the recent study with con results. We have added some comments in Discussion, considering better short- and mid-term survival after surgical management of acute type A intramural hematoma when compared to classic acute type A aortic dissection. (Lines 236 to 244).

Comments 5: Expand the discussion of the results in the context of existing guidelines and consensus documents on the subject.

Response 5: As the reviewer recommend, we have expanded the discussion of the results in the context of existing guidelines. (Lines 199 to 205)

Comments 6: It is recommended to create a graphical abstract to convey the main message of the study effectively.

Response 6: As the reviewer recommend, we have created a graphical abstract. We appreciate your great suggestion to convey our main message effectively.

Comments 7: Highlight any sex-related differences in the outcomes considered.

Response 7: As the reviewer suggested, we recognize the sex difference in outcomes of acute aortic dissection. In this study, we did not observe any sex difference in outcomes of our conservative strategy of AIMH and RT-TAAAD. Since it is an important point, however, we have added some discussion. (Lines 268 to 271).

Reviewer 2 Report

Comments and Suggestions for Authors

I read with great interest the manuscript by Akita et al that evaluated retrospectively 98 patients with aortic intramural hematoma (AIMH) and retrograde thrombosed type A acute aortic dissection (RT-TAAAD), that were treated conservative (78 patients), apart from patients with aortic diameter > 55mm, false lumen expansion or cardiac tamponade that were operated (18 patients). They demonstrated that both early and late outcomes of conservative strategy for AIMH and RT-TAAAD are associated with surgery-free rates.

Author Response

We appreciate that the reviewer read with great interest our manuscript investigating the management of aortic intramural hematoma (AIMH) and retrograde thrombosed type A acute aortic dissection (RT-TAAAD). We would like the reviewer to recognize further that our manuscript will provide a more comprehensive understanding of the conservative strategy's effectiveness in treating AIMH and RT-TAAAD.

Round 2

Reviewer 1 Report

Comments and Suggestions for Authors

In reviewing the revised manuscript, we extend our gratitude to the authors for their diligent attention to the suggestions provided during the previous review cycle. It is evident that the authors have taken our feedback seriously, resulting in a responsive and thoughtful overhaul of the content. Each of the questions raised in the prior round has been addressed comprehensively, showcasing a commendable commitment to enhancing the clarity and coherence of the manuscript. Given the substantive revisions and the enhanced quality of the manuscript, we find it suitable for acceptance in its current form.